# The role of family conflict and cohesion in adolescents' social responsibility: Emotion regulation ability as a mediator

**Wing Yee Cheng[1], Rebecca Y. M. Cheung [2]*, Kevin Kien Hoa Chung[3,4]**

**1** Department of Psychology, The University of Hong Kong, Hong Kong, China, **2** School of Psychology and Clinical Language Sciences, University of Reading, Reading, United Kingdom, **3** Department of Early Childhood Education, The Education University of Hong Kong, Hong Kong, China, **4** Centre for Child and Family Science, The Education University of Hong Kong, Hong Kong, China

* rebecca.cheung@reading.ac.uk

**Data Availability Statement:** Participants' identifying information cannot be shared publicly

## Abstract

The social context is crucial for the adolescent development of self-regulatory skills and social responsibility. To understand the role of social context in adolescent development, the present study examined family predictors (i.e., family cohesion and conflict) of social responsibility, with emotion regulation ability as a mediating process. A total of 828 Chinese adolescents (35.6% female; mean age = 13.92 years, $SD$ = 1.34) were recruited from major Chinese cities, including Hong Kong and Macau. Path analysis revealed that emotion regulation ability mediated the relation between family factors (i.e., family cohesion and family conflict) and social responsibility. That is, the ability to regulate emotions serves as a process between family factors and social responsibility. More specifically, family cohesion was positively associated with emotion regulation ability, whereas family conflict was negatively associated with emotion regulation ability. In turn, emotion regulation ability was positively associated with social responsibility. The results suggested that the family environment and adolescent's emotion regulation ability are important contextual and intrapersonal factors contributing to their development of social responsibility. As an implication, policymakers and practitioners might allocate resources to enrich positive family interactions and cultivate emotional competency to support adolescents' development of social responsibility.

## Introduction

In adolescence, people begin to form a system of beliefs of what a "good citizen" is [1] and consolidate an "altruistic identity" by participating in prosocial services [2], both of which contribute to their development of social responsibility. Social responsibility is a value orientation that reflects people's concern for others extending beyond personal interest [3]. Social responsibility also reflects people's sense of duty to address social needs [4]. For instance, an individual who has a strong sense of social responsibility might have a stronger value for helping those who are less fortunate, working hard to improve society, donating time or money to charities, and engaging in social movements that benefit the country [5]. Such value orientation could

due to ethical concerns. Anonymous data are available in the supplementary material.

**Funding:** The present study was funded by The Education University of Hong Kong and the Centre for Child and Family Science at The Education University of Hong Kong (R3669 & CCFS2017-0003). The funders had no role in study design, data collection and analysis, decision to publish, or preparation of the manuscript.

**Competing interests:** The authors have declared that no competing interests exist.

motive adolescents to engage in more prosocial and civic actions to help build a stronger community [6]. The development of social responsibility in adolescence is beneficial at both personal and societal levels. At the personal level, greater adolescents' social responsibility was found to mediate the relation between in-depth identity exploration and higher civic engagement [4]. Longitudinal studies also found that greater social responsibility was associated with more civic participation in adolescence [7], which further predicted greater social responsibility in young adulthood [7, 8]. Recent studies showed that greater social responsibility was also related to more positive public health behaviors [9] and more civic actions toward social injustice [10] in society. The present study examines the possible roles of family cohesion and family conflict in the development of social responsibility among adolescents to provide insights into promoting positive adolescent development and social change.

## Family cohesion, family conflict, and social responsibility

Family is one of the primary socializing agents whereby adolescents learn how to interact with the world around them [11]. In Chinese culture, individuals highly value family relationships and perceive that the family unit is one of the most basic and important social units [12]. Under the influence of Confucianism in the East Asian context, Chinese families emphasize common good and harmony over personal interests [12], and prefer to avoid conflict and maintain harmonious interparental and parent-child relationships [13, 14]. Together with the socioecological framework [15] and family systems theory [16], the family is postulated to be an important environment for adolescent development, particularly in a Chinese context. According to Bronfenbrenner's socioecological framework [15], parental practices have a direct influence on child and adolescent socioemotional development. As an active agent, children also affect their social environment, creating dynamics in the social ecologies. Similarly, family systems theory [16] describes the family as an organized system composed of interdependent subsystems, such as the interparental subsystem and the parent-child subsystem [6, 15]. The theory highlights the dynamic interaction between family members, generating mutual influences within the family that foster adolescent development [17]. Based on these theoretical frameworks, in this study, we propose that the interplay between family relationships and adolescents could facilitate adolescent development of emotion regulation ability and social responsibility [6, 15].

Supporting the socioecological framework [15] and family systems theory [16], studies to date have found that the indices of the family environment contribute to the development of social responsibility. A positive family environment plays an important role in predicting adolescents' social responsibility [6, 18] and prosocial behaviors [19]. For instance, family cohesion, which refers to the degree of concern and support among family members [20], is related to higher social responsibility [18], more prosocial behaviors [21], and a higher level of community responsibility [22]. Nevertheless, inconsistent findings also exist in the literature to suggest negative or no associations between family cohesion and other indices of social responsibility, such as political activism and volunteerism [23, 24]. The inconsistent findings suggest a need to clarify the mechanisms between family cohesion and social responsibility.

Family conflict indicates the extent of anger, aggression, and conflictual interactions between family members [25]. Conflict can damage families [26] and is destructive to child development. For instance, family conflict could result in poor parent-child relationships that further reinforce adolescents' antisocial behaviors [27, 28]. Previous studies found that the disruption of healthy parent-child relationships was related to adolescents' self-control and resulted in adolescents' problem behaviors [29]. Adolescents who display problem behaviors might be rejected by conventional peers and join deviant peer groups that further reinforce

antisocial behaviors [28]. Although previous research provided insights on the association between family conflict and antisocial behaviors, few studies to our knowledge have directly examined the process between family conflict and social responsibility. Although previous studies did provide preliminary support on the relationship between the family cohesion and the development of social responsibility [18], a research gap remains for the study of different aspects of the family factors (i.e., family cohesion and family conflict) in contributing to adolescents' social responsibility. Given the family environment has a major influence on adolescents' psychosocial development, understanding the relationship between family cohesion, family conflict, and social responsibility is deemed necessary.

Adolescents' social trust might further obscure the link between family factors and social responsibility. Social trust is defined as an individual's belief in the fairness and trustworthiness of their treatment within society [30]. Previous research suggested that family cohesion was positively related to higher social trust among Chinese adolescents [31]. Another study found that family cohesion was related to social responsibility over and above the effect of social trust [18], suggesting that social trust may be linked to family cohesion and social responsibility. Therefore, to identify the unique correlates of social responsibility, we examined the links between family conflict, cohesion, and social responsibility, over and above social trust as a covariate.

## Emotional regulation ability as a mediator

Emotion regulation ability plays an important role among adolescents, who are prone to react strongly to emotions [32]. Emotion regulation is the process of "modulating, evaluating, and modifying emotional reactions, especially their intensive and temporal features, to accomplish one's goals" [33]. There are multiple ways of examining emotion regulation. Previous studies have adopted the process model of emotional regulation [34] and examined emotion regulation strategies, such as attentional deployment, cognitive change, and response modulation. Nevertheless, other studies adopted a different approach and emphasized emotion regulation ability [35, 36], i.e., an individual's ability to regulate their emotions for recovering quickly from psychological distress [37]. Given that emotion regulation ability is associated with adaptation in social relationships and academic performance [38–40], this study examined the role of an overall emotion regulation ability rather than specific emotion regulation strategies.

According to the tripartite model of family impact on the development of emotion regulation [40], family as a social context contributes to adolescents' development of emotion regulation ability through observational learning, modeling, and social referencing [15, 40, 41]. Studies involving children and adolescents indicated that family cohesion and conflict was directly associated with anger regulation [42, 43] and indirectly associated with anger regulation through parental support [44]. In contrast, in a situation where family conflict arises, adolescents' emotional arousal might lower their threshold for subsequent emotional arousals, which might further impair their emotion regulatory ability by heightening their attentional sensitivity to negative emotions [45].

Adolescents' ability to control their own emotions and understand others' emotions could motivate prosocial behaviors [46, 47]. Of note, empirical findings demonstrated that emotion regulation ability was associated with more prosocial behaviors and civic engagement during adolescence [48, 49]. Emotion regulation ability was also a predictor of prosocial behaviors among sixth and tenth graders [48]. In sum, the above studies concluded that adolescents with adaptive emotional regulation ability could regulate their reactions towards situations in a way that lowers personal distress and results in appropriate prosocial reactions, suggesting a possible association between adolescents' emotion regulation ability and social responsibility.

As discussed earlier, the tripartite model of family impact on the development of emotion regulation ability [40] postulated that the family environment is central to emotion regulation ability and psychosocial development among adolescents. For instance, previous research found that self-regulation ability mediated the positive association between parenting and prosocial behaviors [47], thereby suggesting possible links between family cohesion and conflict, adolescents' regulatory ability, and social responsibility [6, 19]. Nonetheless, to the best of our knowledge, little has been done to examine the mediating role of adolescents' emotion regulation ability for the effects of family cohesion and conflict on social responsibility.

## The current study

The present study examined the relationship among family cohesion, family conflict, emotion regulation ability, and social responsibility among adolescents. We hypothesized that family cohesion and family conflict would have a direct effect on social responsibility, and the effects would be mediated by emotional regulation ability. Specifically, we hypothesized that family cohesion would be positively associated with emotion regulation ability, and family conflict would be negatively associated with emotion regulation ability. Emotion regulation ability, in turn, would be positively associated with social responsibility. Taken together, we hypothesized that emotion regulation ability would mediate the effect of family cohesion and family conflicts on social responsibility. Adolescents' gender and age were included covariates, as they were previously found to be associated with emotion regulation ability and prosocial behaviors [50, 51]. Social trust was also examined as a covariate of social responsibility, as its association with social responsibility had been established in previous research [18]. Extending beyond the current theoretical understanding of adolescent development, potential findings could provide initial evidence of emotion regulation as a process for the effects of family cohesion and family conflict on adolescents' social responsibility. Beyond theoretical contributions, the findings could also inform the design and implementation of interventions aimed at promoting adolescents' social responsibility.

## Methods

### Participants

Participants were 828 Chinese adolescents from secondary schools in Hong Kong ($n = 442$) and Macau ($n = 386$). The sample had 35.6% female ($n = 295$), and the age ranged from 11 to 19 years old ($M = 13.92$; $SD = 1.34$). Participants were recruited via school invitations and mass mailing. The study was conducted with the approval of the Ethics Committee of The Education University of Hong Kong. Written informed consent and assent were obtained from the adolescents and their parents prior to their participation in the study. The response rates of recruitment from Hong Kong and Macau were 90.37% and 90.75%, respectively. The median household income per month was HK$20,001–HK$30,000 (approximately US$2,564–US$3,846). 81.92% of mothers and 80.20% of fathers completed high school education. Table 1 summarizes the demographic information. Table 2 shows the means, SDs, and correlations among all variables.

### Measures

**Family cohesion.** The 9-item family cohesion subscale of the Family Environment Scale (FES) [20] measured adolescents' perceived family cohesion. The subscale was rated on a 4-point Likert scale ranging from 1 (*very incorrect*) to 4 (*very correct*). Sample items included, "Family members will really help and support one another" and "There will be a feeling of

**Table 1. Demographic information of the final sample (N = 828).**

| Variable | M (SD) / Number (%) | | |
|---|---|---|---|
| | **Hong Kong** | **Macau** | **Final sample** |
| | **(n = 442)** | **(n = 386)** | **(N = 828)** |
| Adolescents' age | 13.62 (1.15) | 14.29 (1.45) | 13.92 (1.34) |
| Adolescents' gender | | | |
| Female | 223 (50.57%) | 72 (19.30%) | 295 (36.24%) |
| Male | 218 (49.43%) | 301 (80.70%) | 519 (63.76%) |
| Household size | 3.97 (1.07) | 4.16 (1.50) | 4.06 (1.29) |
| Adolescents' number of siblings | 0.91 (0.04) | 1.12 (0.06) | 1.01 (0.99) |
| Household income per month | | | |
| 1. < HK$10,000 | 25 (9.62%) | 10 (5.18%) | 35 (7.73%) |
| 2. HK$10,001–20,000 | 90 (34.62%) | 43 (22.28%) | 133 (29.36%) |
| 3. HK$20,001–30,000 | 68 (26.25%) | 53 (27.46%) | 121 (26.71%) |
| 4. HK$30,001–40,000 | 36 (13.85%) | 43 (22.28%) | 79 (17.44%) |
| 5. HK$40,001–50,000 | 17 (6.54%) | 30 (15.54%) | 47 (10.38%) |
| 6.> HK$50,000 | 24 (9.23%) | 14 (7.25%) | 38 (8.39%) |
| Fathers' level of education | | | |
| 1. Primary school or below | 35 (11.08%) | 54 (19.64%) | 89 (15.06%) |
| 2. Secondary school | 232 (73.42%) | 153 (55.63%) | 385 (65.15%) |
| 3. Bachelor's degree / Higher diploma / Diploma | 37 (11.71%) | 47 (17.09%) | 84 (14.21%) |
| 4. Postgraduate degree or above | 12 (3.80%) | 21 (7.64%) | 33 (5.58%) |
| Mothers' level of education | | | |
| 1. Primary school or below | 46 (14.07%) | 45 (16.30%) | 91 (15.09%) |
| 2. Secondary school | 239 (73.08%) | 164 (59.42%) | 403 (66.83%) |
| 3. Bachelor's degree / Higher diploma / Diploma | 35 (10.70%) | 51 (18.48%) | 86 (14.26%) |
| 4. Postgraduate degree or above | 8 (2.14%) | 16 (5.80%) | 23 (3.81%) |

**Table 2. Means, standard deviations, and correlations among all variables.**

| Variable | (1) | (2) | (3) | (4) | (5) | (6) | (7) | (8) | (9) |
|---|---|---|---|---|---|---|---|---|---|
| (1) Adolescents' gender (0 = male; 1 = female) | — | | | | | | | | |
| (2) Adolescents' age | -.08* | — | | | | | | | |
| (3) Adolescents' number of siblings | -.03 | .04 | — | | | | | | |
| (4) Household income | -.07 | -.09 | .05 | — | | | | | |
| (5) Family cohesion | .02 | -.14*** | -.04 | .19*** | — | | | | |
| (6) Family conflict | -.00 | .12*** | .04 | -.15** | -.58*** | — | | | |
| (7) Adolescents' emotion regulation ability | .04 | -.02 | .05 | .07 | .20*** | -.24*** | — | | |
| (8) Adolescents' social responsibility | .04 | -.06 | .05 | .08 | .22*** | -.08* | .17*** | — | |
| (9) Adolescents' social trust | .00 | -.07* | .00 | -.00 | .23*** | -.19*** | .20*** | .23*** | — |
| M | NA | 13.92 | 1.01 | 3.19 | 2.81 | 2.08 | 4.66 | 3.35 | 3.22 |
| SD | NA | 1.34 | 0.99 | 1.39 | 0.50 | 0.51 | 1.33 | 0.66 | 0.70 |

*Note.* *p = /< .05

**p = /< .01

***p = /< .001. Monthly household income scale ranging from 1 to 6: 1 = less than HK$10,000; 2 = HK$10,001–20,000; 3 = HK$20,001–30,000; 4 = HK$30,001–40,000; 5 = HK$40,001–50,000; 6 = over HK$50,000.

togetherness in the family." A higher average score indicated a higher level of family cohesion. The FES had been validated in Hong Kong for use in a Chinese sample and demonstrated adequate internal consistency, factor structure, and intercorrelations between FES subscales [52]. In this study, the Cronbach's alpha = .77.

**Family conflict.** The 9-item family conflict subscale of the Family Environment Scale (FES) [20] measured adolescents' perceived family conflict. The subscale was rated on a 4-point Likert scale ranging from 1 (*very incorrect*) to 4 (*very correct*). Sample items included, "Family members will sometimes get so angry that they throw things" and "Family members will hardly ever lose their tempers (reversed item)." A higher average score indicated a higher level of family conflict. The FES had been validated in Hong Kong for use in a Chinese sample and demonstrated adequate internal consistency, factor structure, and intercorrelations between FES subscales [52]. In this study, the Cronbach's alpha = .71.

**Emotion regulation ability.** The 4-item subscale of the Emotional Intelligence Scale [37] measured adolescents' emotion regulation ability. The scale was rated on a 7-point Likert scale ranging from 1 (*strongly disagree*) to 7 (*strongly agree*), and higher averaged scores indicated higher levels of emotion regulation ability. Sample items included, "I am able to control my temper and handle difficulties rationally" and "I have good control of my own emotions." In a validation study, the scale demonstrated adequate psychometric properties in Chinese university students and Spanish university students [53, 54]. In this study, the Cronbach's alpha = .90.

**Social responsibility.** The 7-item Social Responsibility Scale [5] measured adolescents' social responsibility. Participants rated the importance of the statement relating to social responsibility on a 5-point Likert scale ranging from 1 (*not at all important*) to 5 (*very important*). Sample items included, "To help those who are less fortunate", "To help others improve their lives", "To work for the betterment of the society", and "To help this country." Higher averaged scores indicated greater social responsibility. Although the scale had not been validated in Chinese or Western samples, a previous study reported that the scale had adequate internal consistency in American adolescents and Chinese adolescents, respectively [18, 55]. In this study, the Cronbach's alpha = .89.

**Social trust.** The 3-item Social Trust Measure [56] assessed the social trust level of adolescents. Participants rated the statements on a 5-point Likert scale ranging from 1 (*strongly disagree*) to 5 (*strongly agree*). Sample items included, "Most people can be trusted", "Most people are fair and don't take advantage of you" and "Most people just look out for themselves rather than try to help others." Higher averaged scores indicated a higher social trust level. No validation studies were found in the Chinese or Western population. In this study, the Cronbach's alpha = .47. Given that the item "Most people just look out for themselves rather than try to help others" contributed to the low alpha, the item was removed in the supplementary analysis. Without the item, the Cronbach's alpha = .84.

**Data analysis.** Prior to data analyses, Little's Missing Completely at Random (MCAR) test was conducted via SPSS to examine the pattern of missing values and determine how the missingness should be treated. Path analysis was conducted via MPLUS, Version 8.3 [57] to investigate the emotion regulation ability as a mediator for the effects of family cohesion and family conflict on adolescents' social responsibility, over and above the covariates identified in previous research (i.e., age, gender, family income, number of siblings, and social trust). Mediation was tested by using 10,000 bootstrapping iterations to obtain the bootstrapped confidence intervals of the indirect effects. The data file in the S1 Appendix includes the variables under study, with a score of "999" denoting a missing value. The MPLUS codes are included in the S1 Appendix. Given the low internal consistency of the three-item Social Trust measure, supplementary path analysis was conducted using a two-item social trust measure that demonstrated a higher internal consistency.

## Results

Participants between the two sites (Hong Kong and Macau) did not differ in variables, including fathers' level of education, family conflict, and social responsibility ($ps > .05$). However, they differed in household income level with $t(451) = 3.20$, $p = .001$, household size with $t(797) = 2.05$, $p = .04$, mothers' level of education with $t(601) = 2.15$, $p = .03$, gender with $\chi^2(1) = 85.48$, $p < .001$), age with $t(810) = 7.29$, $p < .001$, emotion regulation ability (Hong Kong: $M = 4.84$; $SD = 1.25$; Macau: $M = 4.65$; $SD = 1.39$) with $t(821) = -4.29$, $p < .001$, family cohesion (Hong Kong: $M = 2.86$; $SD = .47$; Macau: $M = 2.75$; $SD = .53$) with $t(825) = -3.11$, $p = .002$, and social trust (Hong Kong: $M = 3.34$; $SD = .69$; Macau: $M = 3.08$; $SD = .69$) with $t(823) = -5.38$, $p < .001$. At both sites, socioeconomic indices, including family income and parental education were not significantly associated with the study variables, $ps > .05$.

Regarding the missing data analysis, the MCAR test was not significant, $\chi^2(14) = 22.16$, $p = .075$, indicating that the data were completely missing at random. As such, full information maximum likelihood was used to handle the missing data. The zero-order correlations among all variables are presented in Table 2. Family cohesion was negatively correlated with family conflict ($r = -.58$, $p < .001$) and positively correlated with adolescents' emotion regulation ability ($r = .20$, $p < .001$), social responsibility ($r = .22$, $p < .001$), and social trust ($r = .23$, $p < .001$). Family conflict was negatively correlated with adolescents' emotion regulation ability ($r = -.24$, $p < .001$), social responsibility ($r = -.08$, $p = .02$), and social trust ($r = -.19$, $p < .001$). Adolescents' emotion regulation ability was positively correlated with social responsibility ($r = .17$, $p < .001$) and social trust ($r = .20$, $p < .001$). As for covariates, adolescents' social responsibility was positively correlated with social trust ($r = .23$, $p < .001$). Adolescents' age was negatively correlated with family cohesion ($r = -.14$, $p < .001$) and social trust ($r = -.07$, $p = .04$) and positively correlated with family conflict ($r = .12$, $p < .001$). Household income was positively correlated with family cohesion ($r = .19$, $p < .001$) and negatively correlated with family conflict ($r = -.15$, $p = .002$).

The path model fit adequately to the data, $\chi^2(10) = 18.81$, $p = .04$, CFI = .94, TLI = .92, RMSEA = .03, SRMR = .02 (see Fig 1 and Table 3). In the model, family cohesion was positively associated with emotion regulation ability ($\beta = .58$, $p = .003$), and family conflict was negatively associated with emotion regulation ability ($\beta = -.40$, $p = .02$). Emotion regulation ability, in turn, was related to social responsibility ($\beta = .11$, $p = .001$). Adolescents' age was

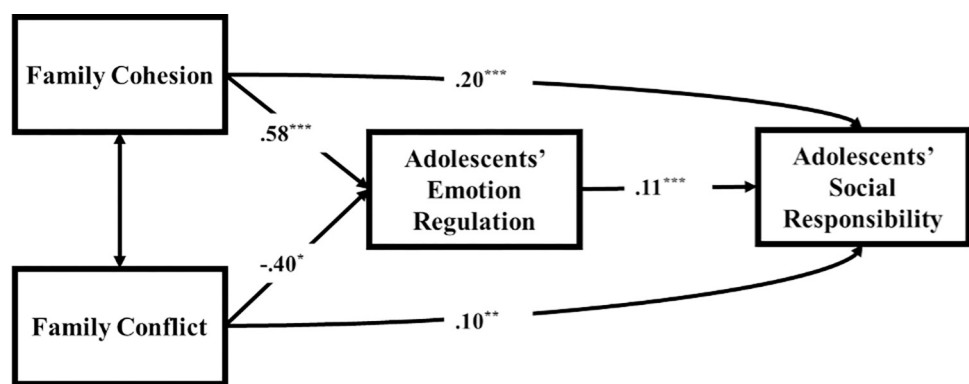

**Fig 1. Adolescents' emotion regulation ability as a mediator between family cohesion, family conflict, and adolescents' social responsibility.** $\chi^2(10) = 18.81$, $p = .04$, CFI = .94, TLI = .92, RMSEA = .03, SRMR = .02. Standardized parameter estimates are presented. Adolescents' social trust, age, gender, number of siblings, and family income were included as covariates but are not depicted in the figure for clarity. *$p = /< .05$, **$p = /< .01$, ***$p = /< .001$.

**Table 3. Parameter estimates of the path model.**

| Parameter | Unstandardized B (SE) | Standardized β |
|---|---|---|
| **Path Model** | | |
| Family cohesion | | |
| → Adolescents' emotion regulation ability | 1.53 (.52) | .58** |
| → Adolescents' social responsibility | .27 (.06) | .20*** |
| Family conflict | | |
| → Adolescents' emotion regulation ability | -1.06 (.47) | -.40* |
| → Adolescents' social responsibility | .14 (.05) | .10** |
| Adolescents' emotion regulation ability | | |
| → Adolescents' social responsibility | .06 (.02) | .11*** |
| Adolescents' gender (0 = male; 1 = female) | | |
| → Adolescents' emotion regulation ability | .10 (.11) | .04 |
| → Adolescents' social responsibility | .05 (.05) | .03 |
| Adolescents' age | | |
| → Adolescents' emotion regulation ability | .10 (.05) | .10* |
| → Adolescents' social responsibility | -.01 (.02) | -.02 |
| Adolescents' number of siblings | | |
| → Adolescents' emotion regulation ability | .11 (.06) | .08* |
| → Adolescents' social responsibility | .03 (.02) | .05 |
| Household income | | |
| → Adolescents' emotion regulation ability | -2.99 (1.04) | -3.11** |
| → Adolescents' social responsibility | .03 (.02) | .06 |
| Covariates | | |
| Family cohesion | | |
| ←→ Family conflict | -.15 (.01) | -.58*** |
| ←→ Adolescents' social trust | .08 (.01) | .22*** |
| ←→ Adolescents' age | -.08 (.02) | -.11*** |
| ←→ Household income | .13 (.03) | .19*** |
| Family conflict | | |
| ←→ Adolescents' age | .06 (.02) | .09** |
| ←→ Household income | -.11 (.03) | -.15*** |
| Adolescents' emotion regulation ability | | |
| ←→ Household income | 5.53 (1.95) | .93*** |
| Adolescents' social trust | | |
| ←→ Adolescents' social responsibility | .17 (.03) | .18*** |

*Note.* *$p = / < .05$

**$p = / < .01$

***$p = / < .001$.

negatively associated with family cohesion ($β = -.11$, $p = .001$), positively associated with family conflict ($β = .09$, $p = .007$), and not associated with other variables ($ps > .05$). Adolescents' gender was not related to all variables under study ($ps > .05$). Family income was positively associated with family cohesion ($β = .19$, $p < .001$) and emotion regulation ability ($β = .93$, $p < .001$), negatively associated with family conflict ($β = -.15$, $p = .001$), and not associated with other variables ($ps > .05$). The number of siblings was positively associated with emotion regulation ability ($β = .08$, $p = .04$) and not with other variables ($ps > .05$). Social trust was positively related to social responsibility ($β = .18$, $p < .001$).

The indirect effects of family cohesion and family conflict on social responsibility via emotion regulation ability were significant (family cohesion: β = .06, *p* = .03; family conflict: β = -.05, *p* = .05). Based on 10000 bootstrap samples with replacement, the 95% confidence interval (CI) indicated that the standardized indirect effect from family cohesion and family conflict to adolescents' social responsibility via emotion regulation ability did not include a zero [β = .06, CI: (.03, .22) and β = -.05 (-.11, .01), respectively], thereby suggesting emotion regulation ability as a mediator.

## Supplementary analysis with a two-item measure of social trust

Supplementary analysis was conducted with a two-item measure of social trust that exhibited a greater internal consistency. However, the mediation model replaced by the two-item measure of social trust did not converge, suggesting potential issues with the computations. As such, predictions were not made in the supplementary analysis.

## Discussion

This research examined the mediating role of emotion regulation ability between the indices of family environment (i.e., family cohesion and family conflict) and social responsibility among Chinese adolescents. The initial findings generally supported the mediation model, thereby supporting the idea that contextual and intrapersonal factors shape adolescents' development of social responsibility [6] and enhancing the current understanding of social responsibility from family systems and ecological perspectives [15, 16].

From a family systems perspective, our measures of the family environment, namely family cohesion and family conflict, reflect the mutual influence of the multiple subsystems in the family (inclusive of a couple, parent-child, and sibling relationships) [17]. Given that existing research primarily reported only one aspect of family environment, e.g., either family cohesion or family conflict [18, 21, 22], this study suggested a more comprehensive view by including family cohesion as a positive index and family conflict as a negative index of the family environment. We found that higher levels of family cohesion and family conflict were both significantly associated with adolescents' higher levels of social responsibility. The finding on the positive relationship between family cohesion and adolescents' social responsibility was consistent with previous studies [18, 21]. As for the surprising positive association between family conflict and social responsibility, we speculate that adolescents with higher family conflict might have sought recognition outside their families, such as from peers and social groups, which could motivate them to develop a stronger sense of social responsibility [58]. As such, future studies should include third variables, such as peer and social support, to understand the relation between family conflict and social responsibility.

In addition to the associations between family cohesion, family conflict, and the development of social responsibility, our study also demonstrated the mediating role of adolescents' emotion regulation ability, thereby supporting previous research in this area [32, 40]. Specifically, our results showed that family cohesion was associated with adolescents' better emotion regulation ability, whereas family conflict was associated with poorer emotion regulation ability. Consistent with the socioecological framework [15], which states that development occurs as a result of regular exchanges between an individual and the context [15], the findings suggested that the family context is crucial in fostering adolescents' emotion regulation ability. Previous studies suggested that parent-child interactions provide modeling and social referencing opportunities for adolescents to develop their emotion regulatory ability, such as how they understand emotions and how to manage their emotions [40, 41, 59]. Thus, a positive family environment with higher family cohesion and lower family conflict could

potentially foster adolescent development of emotion regulatory ability. Our study further indicated that adolescents' emotion regulation ability was associated with greater social responsibility, which is consistent with previous research [48, 49]. As such, the finding under-scores the importance of adolescents' ability to manage and understand emotions in fostering a sense of social responsibility, such as prioritizing their sense of duty towards group or social needs over personal interest.

Taken together, this study revealed the mediating role of emotion regulation ability in the relationship between family cohesion, family conflict, and adolescents' social responsibility, suggesting emotion regulation ability as an internal mechanism underlying the association between family environment and the development of social responsibility. While previous studies had concluded a solid ground for the importance of the external environment on the development of social responsibility (e.g., family compassion, school solidarity, community connectedness, and trusted friendship predicted changes in social responsibility value [58]), our findings added new insights into the role of an intrapersonal factor (i.e., adolescents' emotion regulation ability) of social responsibility. In addition, the study was conducted in a Chinese cultural context that emphasizes the importance of family harmony and cohesion as a foundation of the society [12]. We speculated that this cultural context, which values a high-cohesion and low-conflict family environment, might have particularly contributed to cultivating adolescents' emotion regulation ability and social responsibility. To examine the strength of associations among the variables across cultures, cross-cultural studies between Eastern and Western contexts are necessary in the future.

The present findings have implications for fostering adolescents' social responsibility. Importantly, they suggest that a family environment characterized by higher family cohesion and lower family conflict may enhance adolescents' emotion regulation ability, which in turn promotes social responsibility. These findings have implications for applied research to involve caregivers, educators, policymakers, and other practitioners in cultivating adolescents' emotion regulation ability and social responsibility.

## Limitations and future directions

The limitations of this study should be noted in addition to future directions. First, to clarify the temporal relationships between these variables, future research could adopt a longitudinal framework. Second, the variables under study were self-reported, thereby introducing self-report bias revolving around social desirability and a potential lack of authenticity in the reports. Future studies should, therefore, use a multi-method, multi-informant approach to minimize the potential biases introduced in the study. Third, the Social Trust Measure [56] had a low Cronbach's alpha at .47, which was a major limitation of this study. Pallant [60] mentioned that measures with fewer than ten items could have low Cronbach's alphas, which might be the reason behind the low alpha of social trust in our sample. Upon deletion of the item contributing to the low alpha, we conducted additional analysis for this study. However, predictions could not be made as the path model did not converge. Given the poor internal consistency of the Social Trust Measure [56], validation studies should be conducted to ensure the measure is valid for use in Chinese samples. Cautions should also be made in interpreting the present findings and replication studies should be conducted in the future. Similarly, the Social Responsibility Scale has not been validated in the Chinese context. Future research should conduct validation studies to assess its factor structure and validity. Fourth, this study did not include multiple intrapersonal variables such as empathy, which might be a possible covariate [46, 61]. Similarly, other contextual variables such as school, peer relationships, and neighborhood were not measured. As previous studies indicated that these contextual factors

were important to the development of emotion regulation ability [45] and social responsibility [6], future research could explore how contextual variables might interact with emotion regulation ability in cultivating adolescents' social responsibility. Lastly, although the current study discussed the role of Chinese culture on family dynamics, we did not conduct a cross-cultural study or include cultural values as variables under study. Researchers should further examine the role of cultural values in family dynamics and adolescent development, particularly across cultural contexts.

Despite the above limitations, our initial findings supported the mediating role of emotion regulation ability for the relationship between indices of family environment (i.e., family cohesion and family conflict) and adolescents' social responsibility. The results provided insights for educators and parents to understand the development of adolescents' social responsibility through contextual and intrapersonal factors, including the immediate environment and adolescents' psychological functioning such as emotion regulation ability. Based on the present findings, educators and policymakers may consider allocating resources to enrich positive family interactions and cultivate emotional competency and adolescent social responsibility.

## Supporting information

**S1 Dataset.**
(CSV)

**S1 Appendix.**
(DOCX)

## Acknowledgments

We would like to thank Dr Hoi Yan Cheung for her support in data collection.

## Author Contributions

**Conceptualization:** Rebecca Y. M. Cheung.

**Data curation:** Rebecca Y. M. Cheung.

**Formal analysis:** Wing Yee Cheng.

**Funding acquisition:** Kevin Kien Hoa Chung.

**Investigation:** Wing Yee Cheng, Rebecca Y. M. Cheung, Kevin Kien Hoa Chung.

**Methodology:** Rebecca Y. M. Cheung.

**Project administration:** Wing Yee Cheng, Rebecca Y. M. Cheung.

**Supervision:** Rebecca Y. M. Cheung.

**Writing – original draft:** Wing Yee Cheng, Rebecca Y. M. Cheung.

**Writing – review & editing:** Wing Yee Cheng, Rebecca Y. M. Cheung, Kevin Kien Hoa Chung.

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
