## [Decision Letter · Decision Letter 0]

8 Aug 2024

PONE-D-24-17148Family Processes and Adolescents’ Social Responsibility: The Mediating Role of Emotion RegulationPLOS ONE

Dear Dr. Cheung,

Thank you for submitting your manuscript to PLOS ONE. After careful consideration, we feel that it has merit but does not fully meet PLOS ONE’s publication criteria as it currently stands. Therefore, we invite you to submit a revised version of the manuscript that addresses the points raised during the review process.

Editor's comments:

We apologize for the delay in sending out the decision. This delay was due to difficulties in finding potential reviewers. 

Although we were only able to find one reviewer, they provided useful comments on your manuscript and I also agreed with these comments.

I also reviewed your manuscript throughout and confirmed that it was well-written and in accordance with PLOS ONE's publication policy (i.e., there were no major problems for scientific research). 

The methodology and analysis are brief, but the findings and Chinese data may have important implications for the research context of family psychology.

As the reviewer pointed out, the sections about data analysis and discussion should be improved.

Hypotheses should be written clearly. 

The data analysis section is too short. Although the data file is available, the analysis code should also be available to improve reproducibility. A more detailed supplementary explanation for how to read the data file (e.g., "999" means a missing value) would also be better.

You should discuss the results regarding the reason for the positive relationship between family conflict and social responsibility, although this may be irrelevant to the hypothesis. 

Please check the reviewer's comments and revise your manuscript according to the comments with point-by-point responses.

We look forward to receiving your revised manuscript.

Kind regards,

Yutaka Horita

Academic Editor

PLOS ONE

Journal Requirements:

"The present study was funded by The Education University of Hong Kong and the Centre for Child and Family Science at The Education University of Hong Kong (R3669 & CCFS2017-0003). "

Reviewers' comments:

Reviewer's Responses to Questions

**Comments to the Author**

1. Is the manuscript technically sound, and do the data support the conclusions?

Reviewer #1: Yes

2. Has the statistical analysis been performed appropriately and rigorously? 

Reviewer #1: Yes

3. Have the authors made all data underlying the findings in their manuscript fully available?

Reviewer #1: Yes

4. Is the manuscript presented in an intelligible fashion and written in standard English?

Reviewer #1: Yes

5. Review Comments to the Author

Reviewer #1: The manuscript is written in a clear and coherent way and is based on data collected from a large sample. however, there are several issues that need to be addressed.

The title is misleading. It is hard to understand what “family process” refers to.

The abstract is not informative enough. The authors have tested a mediation model. It is hard to understand this by reading the abstract.

Introduction

The following sentence can be revised “As another index of the family environment, family conflict indicates the extent of anger, aggression, and conflictual interactions between family members”.

The authors state that the study is based on family systems theory and the ecological framework. However, they paid little attention to these models. The introduction should be reorganized by putting more emphasis on its theoretical basis.

The concept “social responsibility” needs to be clearly explained.

The following sentence is difficult to follow “The relationship between family conflict and emotion regulation was also examined with impulsivity (34), and the findings suggested that ongoing family conflict was related to an increased risk for greater emotional negativity and greater aggression over time (34, 35)”.

The section on the mediator role of ER can be reorganized by making sure that the authors cover only the literature directly related to the current study. For example, the have cited studies that focus on empathy which is indirectly related to the current research question. Also, the study that I have mentioned above on impulsivity seems indirectly related.

Although the authors have provided a definition of ER. However, an examination of previous studies indicated that there are multiple ways of examining ER. Certain studies approach ER based on the use of specific strategies. Some others focus on difficulties in ER or ER flexibility. It is thus, important to explain the specific ER framework utilized in the present study.

Family process, family cohesion, family precursor are different names used with reference to the same concept. It may be a good idea to be consistent while using terms.

At the end of the introduction the authors stated, “Social trust was also examined as the covariate of social responsibility”. This concept appears for the first time in the final sentence of the introduction section. The authors have provided a justification. However, the justification is not enough to explain why the authors have included social trust as a covariate. More explanation is needed.

The hypotheses should be written more clearly, I did not understand that both family cohesion and family conflict act as the independent variables until I started reading the results section.

Also the final paragraph of introduction should give certain cues regarding the potential contribution of the present study to the literature.

Method

The participants section is well-written and informative. However, I have concerns regarding the presentation of t-tests and chi-square results in the participants section rather than the results section.

The authors, I assume, used the Chinese versions of the scales mentioned. It is crucial to report the psychometric properties of the Chinese version of the scales. Are they culturally appropriate, were there any problems in the adaptation process etc.

It may be a good idea to use the term “emotion regulation ability” rather than just emotion regulation”.

Also, the authors have used quite brief versions of the scales. I wonder whether they are efficiently assessing the constructs that form the study variables.

Data analysis section should be more detailed.

Results

Correlation results are provided via the Table 2, but there is no section in the results section explain the results of correlation analyses.

The following paragraph can be written in a better way “The indirect effects of family cohesion and family conflict to social responsibility via emotion regulation were significant (β = .03, p < .01). Based on 10000 bootstrap samples with replacement, the 95% confidence interval (CI) indicated that the standardized indirect effect from family cohesion and family conflict to adolescents’ social responsibility via emotion regulation did not include a zero [CI: (.03, .19) for family cohesion and (-.14, -.01) for family conflict], thereby suggesting emotion regulation as a mediator.” Coefficients should be added.

Discussion

There is some repetition in the discussion section. The first two paragraphs highlight the same issues. The authors should write more concisely.

The mediation results should be discussed more thoroughly.

A Cronbach’s alpha coefficient of .47 is really low. It is a significant limitation that needs to be addressed more clearly. The authors can check the items and look for the item that is responsible for the low internal consistency. In line with one of my previous comments, it can be useful to check the reliability coefficients of the original version and the first Chinese adaptation of the scale as well.

Have the authors checked the participant authenticity?

The data were collected from Chinese adolescents. It would be interesting to read something about the possible impact of culture at least in the discussion section, keeping in mind that this is not a cross-cultural study.

6. PLOS authors have the option to publish the peer review history of their article (what does this mean?). If published, this will include your full peer review and any attached files.

Reviewer #1: No

---

## [Author Response · Author response to Decision Letter 0]

29 Aug 2024

Editor’s comments:

• The methodology and analysis are brief, but the findings and Chinese data may have important implications for the research context of family psychology.

• Thank you for your comment. We have now provided further details in the methodology and analysis to enrich the manuscript.

• As the reviewer pointed out, the sections about data analysis and discussion should be improved.

• We have revised the data analysis section and the discussion according to the reviewer’s recommendation. Please see our responses below for the details. Thank you.

• Hypotheses should be written clearly. 

• Thank you for your comment. We have revised the hypotheses in the manuscript as follows, “We hypothesized that family cohesion and family conflict would have a direct effect on social responsibility, and the effects would be mediated by emotional regulation ability. Specifically, we hypothesized that family cohesion would be positively associated with emotion regulation ability, and family conflict would be negatively associated with emotion regulation ability. Emotion regulation ability, in turn, would be positively associated with social responsibility. Taken together, we hypothesized that emotion regulation ability would mediate the effect of family cohesion and family conflicts on social responsibility.” (lines 131-138)

• The data analysis section is too short. Although the data file is available, the analysis code should also be available to improve reproducibility. A more detailed supplementary explanation for how to read the data file (e.g., "999" means a missing value) would also be better.

• Thank you for the comment. We have revised the data analysis section (lines 214-216), “Prior to data analyses, Little’s Missing Completely at Random (MCAR) test was conducted via SPSS to examine the pattern of missing values and determine how the missingness should be treated.” In the results section, we have added (lines 236-238), “Regarding the missing data analysis, the MCAR test was not significant, χ2(14) = 22.16, p = .075, indicating that the data were completely missing at random. As such, full information maximum likelihood was used to handle the missing data.”

To enable the readers to understand the analysis and read the data file, we have added the following in the manuscript (lines 216-222), “Path analysis was conducted via MPLUS, Version 8.3 [58] to investigate the emotion regulation ability as a mediator for the effects of family cohesion and family conflict on adolescents’ social responsibility, over and above the covariates identified in previous research (i.e., age, gender, family income, number of siblings, and social trust). Mediation was tested by using 10,000 bootstrapping iterations to obtain the bootstrapped confidence intervals of the indirect effects. The data file in the Appendix includes the variables under study, with a score of “999” denotes a missing value. The MPLUS codes are included in the appendix.” 

• You should discuss the results regarding the reason for the positive relationship between family conflict and social responsibility, although this may be irrelevant to the hypothesis. 

• Thank you for your comment. We added the discussion on this finding, “We found that higher levels of family cohesion and family conflict were both significantly associated with adolescents’ higher level of social responsibility. The finding on the positive relationship between family cohesion and adolescents’ social responsibility was consistent with previous studies [18, 21]. As for the surprising positive association between family conflict and social responsibility, we speculate that adolescents with higher family conflict might have sought recognition outside their families, such as from peers and social groups, which could motivate them to develop a stronger sense of social responsibility [59]. As such, future studies should include third variables, such as peer and social support, to understand the relation between family conflict and social responsibility.” (lines 299-307)

• Thank you for the reminders. We have reformatted the manuscript to meet the requirements.

• Thank you for stating the following financial disclosure: 

“The present study was funded by The Education University of Hong Kong and the Centre for Child and Family Science at The Education University of Hong Kong (R3669 & CCFS2017-0003).” Please state what role the funders took in the study. If the funders had no role, please state: “The funders had no role in study design, data collection and analysis, decision to publish, or preparation of the manuscript.” 

• Thank you for the reminders. In the cover letter, we have now stated, “The funders had no role in study design, data collection and analysis, decision to publish, or preparation of the manuscript.” 

• Please include your full ethics statement in the ‘Methods’ section of your manuscript file. In your statement, please include the full name of the IRB or ethics committee who approved or waived your study, as well as whether or not you obtained informed written or verbal consent. If consent was waived for your study, please include this information in your statement as well. 

• The full ethics statement has been included in the ‘Methods’ section of the manuscript file, “The study was conducted with the approval of the Ethics Committee of The Education University of Hong Kong. Written informed consent and assent were obtained from the adolescents and their parents prior to their participation in the study.” (lines 153-155).

• Thank you for your reminder. We have reviewed the reference list.

Reviewer’s comments:

• The manuscript is written in a clear and coherent way and is based on data collected from a large sample. 

• Thank you for the positive comment.

• The title is misleading. It is hard to understand what “family process” refers to.

• Thank you for your comment. We revised the title to, “The Role of Family Conflict and Cohesion in Adolescents’ Social Responsibility: Emotion Regulation Ability as a Mediator”

• The abstract is not informative enough. The authors have tested a mediation model. It is hard to understand this by reading the abstract.

• Thank you for the comment. The abstract has been revised to emphasize we have tested a mediation model. It reads (lines 3-5), “To understand the role of social context in adolescent development, the present study examined family predictors (i.e., family cohesion and conflict) of social responsibility, with emotion regulation ability as a mediating process.”

• Introduction - The following sentence can be revised “As another index of the family environment, family conflict indicates the extent of anger, aggression, and conflictual interactions between family members”.

• Thank you for the comment. We deleted “As another index of the family environment” to avoid confusion. It now reads (lines 67-68), “Family conflict indicates the extent of anger, aggression, and conflictual interactions between family members [25].”

• The authors state that the study is based on family systems theory and the ecological framework. However, they paid little attention to these models. The introduction should be reorganized by putting more emphasis on its theoretical basis.

• Thank you for your comment. We have now elaborated on the theoretical frameworks in the Introduction: “Together with the socioecological framework [15] and family systems theory [16], family is postulated to be an important environment for adolescent development, particularly in a Chinese context. According to Bronfenbrenner’s socioecological framework [15], parental practices have a direct influence on child and adolescent socioemotional development. As an active agent, children also affect their social environment, creating dynamics in the social ecologies. Similarly, family systems theory [16] describes the family as an organized system composed of interdependent subsystems such as the interparental subsystem and the parent-child subsystem [6, 15]. The theory highlights the dynamic interaction between family members, generating mutual influences within the family that foster adolescent development [17]. Based on these theoretical frameworks, in this study we propose that the interplay between family relationships and adolescents could facilitate adolescent development of emotion regulation ability and social responsibility [6, 15]. ” (lines 45-56)

• The concept “social responsibility” needs to be clearly explained.

• We have now defined the concept of social responsibility in the first paragraph, “Social responsibility is a value orientation that reflects people’s concern for others extending beyond personal interest [3]. Social responsibility also reflects people’s sense of duty to address social needs [4]. For instance, an individual who has a strong sense of social responsibility might have a stronger value for helping those who are less fortunate, working hard to improve society, donating time or money to charities, and engaging in social movements that benefit the country [5].” (lines 22-27)

• The following sentence is difficult to follow “The relationship between family conflict and emotion regulation was also examined with impulsivity (34), and the findings suggested that ongoing family conflict was related to an increased risk for greater emotional negativity and greater aggression over time (34, 35)”.

• Thank you for your comment. We have deleted the sentence, “The relationship between family conflict and emotion regulation was also examined with impulsivity (34), and the findings suggested that ongoing family conflict was related to an increased risk for greater emotional negativity and greater aggression over time (34, 35)”. to avoid confusion, as they were not directly related to our research question.

• The section on the mediator role of ER can be reorganized by making sure that the authors cover only the literature directly related to the current study. For example, the have cited studies that focus on empathy which is indirectly related to the current research question. Also, the study that I have mentioned above on impulsivity seems indirectly related.

• Thank you for the suggestion. We have trimmed down the studies that were not directly related to the research question to avoid confusion, including studies revolving around empathy and impulsivity.

• Although the authors have provided a definition of ER. However, an examination of previous studies indicated that there are multiple ways of examining ER. Certain studies approach ER based on the use of specific strategies. Some others focus on difficulties in ER or ER flexibility. It is thus, important to explain the specific ER framework utilized in the present study.

• Thank you for your comment. We clarified the definition of ER in the present study, “There are multiple ways of examining emotion regulation. Previous studies have adopted the process model of emotional regulation [34] and examined emotion regulation strategies, such as attentional deployment, cognitive change, and response modulation. Nevertheless, other studies adopted a different approach and emphasized emotion regulation ability [35, 36], i.e., an individual’s ability to regulate their emotions for recovering quickly from psychological distress [37]. Given that emotion regulation ability is associated with adaptation in social relationships and academic performance [38-40], this study examined the role of an overall emotion regulation ability, rather than specific emotion regulation strategies.” (lines 96-103)

• Family process, family cohesion, family precursor are different names used with reference to the same concept. It may be a good idea to be consistent while using terms.

• Thank you for your suggestion. We have now minimized the use of multiple terms to increase consistency.

• At the end of the introduction the authors stated, “Social trust was also examined as the covariate of social responsibility”. This concept appears for the first time in the final sentence of the introduction section. The authors have provided a justification. However, the justification is not enough to explain why the authors have included social trust as a covariate. More explanation is needed.

• Thank you for your suggestion. We have elaborated on social trust in the Introduction, “Adolescents’ social trust might further obscure the link between family factors and social responsibility. Social trust is defined as an individual’s belief in the fairness and trustworthiness of their treatment within the society [30]. Previous research suggested that family cohesion was positively related to higher social trust among Chinese adolescents [31]. Another study found that family cohesion was related to social responsibility over and above the effect of social trust [18], suggesting that social trust may be linked to family cohesion and social responsibility. Therefore, to identify the unique correlates of social responsibility, we examined the links between family conflict, cohesion, and social responsibility, over and above social trust as covariate.” (lines 83-91)

• The hypotheses should be written more clearly, I did not understand that both family cohesion and family conflict act as the independent variables until I started reading the results section.

• Thank you for your comment. The hypotheses have been rewritten, “We hypothesized that family cohesion and family conflict would have a direct effect on social responsibility, and the effects would be mediated by emotional regulation ability. Specifically, we hypothesized that family cohesion would be positively associated with emotion regulation ability, and family conflict would be negatively associated with emotion regulation ability. Emotion regulation ability, in turn, would be positively associated with social responsibility. Taken together, we hypothesized that emotion regulation ability would mediate the effect of family cohesion and family conflicts on social responsibility.” (lines 131-138) 

• The final paragraph of introduction should give certain cues regarding the potential contribution of the present study to the literature.

• Thank you for your comment. We have added the potential contribution of the present study, “Extending beyond the current theoretical understanding of adolescent development, potential findings could provide initial evidence of emotion regulation as a process for the effects of family cohesion and family conflict on adolescents’ social responsibility. Beyond theoretical contributions, the findings could also inform the design and implementation of interventions aimed at promoting adolescents’ social responsibility.” (lines 141-146)

• Method - The participants section is well-written and informative. However, I have concerns regarding the presentation of t-tests and chi-square results in the participants section rather than the results section.

• Thank you for the comment. We have moved the presentation of t-tests and chi-square results to the Results section (lines 226-235).

• The authors, I assume, used the Chinese versions of the scales mentioned. It is crucial to report the psychometric properties of the Chinese version of the scales. Are they culturally appropriate, were there any problems in the adaptation process

---

## [Editor Report · Decision Letter 1]

13 Sep 2024

PONE-D-24-17148R1The Role of Family Conflict and Cohesion in Adolescents’ Social Responsibility: Emotion Regulation Ability as a MediatorPLOS ONE

Dear Dr. Cheung,

Thank you for submitting your manuscript to PLOS ONE. After careful consideration, we feel that it has merit but does not fully meet PLOS ONE’s publication criteria as it currently stands. Therefore, we invite you to submit a revised version of the manuscript that addresses the points raised during the review process.

Academic editor:I have checked the revised manuscript and believe that the issues have been addressed. However, let me point out a minor issue about data. 

In the revised manuscript, you described that "The MPLUS codes are included in the appendix (line 222)". However, although you attached the data file (dataset.csv), I could not find the code file. Please attach the code as supplementary information or let readers know how to access it (e.g., url to access the online repository).

In addition, in the data availability statement you attached, you stated that data is available upon a request. This statement is also inconsistent with the main text. I believe that the descriptions about data availability have recently been important for reproducibility.  I hope that you correct these inconsistent descriptions.

https://journals.plos.org/plosone/s/data-availability

We look forward to receiving your revised manuscript.

Kind regards,

Yutaka Horita

Academic Editor

PLOS ONE
---

## [Author Response · Author response to Decision Letter 1]

13 Sep 2024

13 September 2024

Dear Editor(s),

We are resubmitting our revised manuscript entitled “Family Processes and Adolescents’ Social Responsibility: The Mediating Role of Emotion Regulation” (PONE-D-24-17148R1) for review and possible publication in PLOS ONE. We thank you for the careful review and comments. In this submission, we have included the MPLUS codes and an updated data availability statement. For your easy reference, the data availability statement has been highlighted in yellow.

We affirm that the study was conducted in accordance with the ethical standards of the American Psychological Association. The findings have not been published previously. The present study was funded by The Education University of Hong Kong and the Centre for Child and Family Science at The Education University of Hong Kong (R3669 & CCFS2017-0003). The funders had no role in study design, data collection and analysis, decision to publish, or preparation of the manuscript.

We confirm that the order of authorship of this manuscript corresponds to the authors’ relative contributions to the research effort reported in the manuscript. Thank you very much for your consideration of our manuscript at PLOS ONE. We look forward to your decision in due course.

Sincerely,

Rebecca Y. M. Cheung, Ph.D.

University of Reading

---

## [Editor Report · Decision Letter 2]

17 Sep 2024

The Role of Family Conflict and Cohesion in Adolescents’ Social Responsibility: Emotion Regulation Ability as a Mediator

PONE-D-24-17148R2

Dear Dr. Cheung,

We’re pleased to inform you that your manuscript has been judged scientifically suitable for publication and will be formally accepted for publication once it meets all outstanding technical requirements.

Kind regards,

Yutaka Horita

Academic Editor

PLOS ONE
---

## [Editor Report · Acceptance letter]

20 Sep 2024

PONE-D-24-17148R2 

PLOS ONE

Dear Dr. Cheung, 

I'm pleased to inform you that your manuscript has been deemed suitable for publication in PLOS ONE. Congratulations! Your manuscript is now being handed over to our production team.

Kind regards, 

on behalf of

Dr. Yutaka Horita 

Academic Editor

PLOS ONE